# Prognostic Factors Associated with Recovery from Recurrent Idiopathic Sudden Sensorineural Hearing Loss: Retrospective Analysis and Systematic Review

**DOI:** 10.3390/jcm11051453

**Published:** 2022-03-07

**Authors:** So Young Jeon, Dae Woong Kang, Sang Hoon Kim, Jae Yong Byun, Seung Geun Yeo

**Affiliations:** Department of Otorhinolaryngology-Head & Neck Surgery, School of Medicine, Kyung Hee University, 23 Kyungheedae-ro, Dongdaemun-gu, Seoul 02447, Korea; iamjsy89@gmail.com (S.Y.J.); kkang814@naver.com (D.W.K.); hoon0700@naver.com (S.H.K.); otorhioo512@naver.com (J.Y.B.)

**Keywords:** sudden, idiopathic sensorineural hearing loss, recurrent

## Abstract

Although idiopathic sudden sensorineural hearing loss (ISSNHL) is uncommon, recurrent ISSNHL is even rarer. The knowledge about factors associated with patient recovery from recurrent episodes is needed to counsel and treat the patients. Medical records of patients admitted for high dose oral steroid therapy for recurrent ISSNHL between January 2009 and December 2021 were reviewed. Their demographic and clinical characteristics, co-morbid symptoms, and audiologic results were analyzed. The 38 patients admitted for treatment of recurrent ISSNHL included 14 men and 24 women. Recovery rates after the first and recurrent episodes of ISSNHL were 78.9% and 63.2%, respectively. Patients who recovered after recurrent episodes showed significantly higher rates of ear fullness symptoms and early treatment onset than those who did not recover (*p* < 0.05 each). Of the 30 patients who recovered after the first episode, those who had ear fullness symptoms (*p* < 0.05, odds ratio (OR) 0.1, 95% confidence interval (CI) 0.01–0.76) and who showed a lower initial hearing threshold (*p* < 0.05, OR 1.06, 95% CI 1.01–1.12) during the recurrent episode showed significantly better or similar recovery than after the first episode. Ear fullness symptoms and less initial hearing loss were associated with a more favorable prognosis after intial than after recurrent ISSNHL.

## 1. Introduction

Idiopathic sudden sensorineural hearing loss (ISSNHL) is typically defined as a sensorineural hearing loss greater than 30 dB over at least three consecutive frequencies within three days [1]. Its incidence is 5 to 20 per 100,000 people per year [2]. Studies analyzing long-term outcomes of ISSNHL have found that recurrence rates vary widely, ranging between 0.8% and 47% [3,4,5,6,7,8].

There is no known single cause of ISSNHL, but factors may include viral infection, vascular disturbance, cochlear membrane rupture, immune-mediated mechanisms, tumor, autonomic nervous disease, ototoxic drugs, and congenital anomaly, or it may be idiopathic [9]. Although many studies have assessed the causes, pathogenesis, diagnostic criteria, treatment, and prognostic factors of ISSNHL [10,11,12], few have analyzed the clinical features and prognostic factors associated with recurrent ISSNHL. Furthermore, to our knowledge, no study has compared the clinical features or factors associated with recovery from initial and recurrent episodes of ISSNHL. This study retrospectively analyzed 38 patients with recurrent ISSNHL and compared the clinical features and prognostic factors associated with recovery from initial and recurrent episodes of ISSNHL.

## 2. Subjects and Methods

### 2.1. Study Design

This retrospective study included patients with recurrent ISSNHL who visited the ear, nose, and throat (ENT) clinic at Kyung-Hee Medical Center and Gang-dong Kyung-Hee Medical Center for SSNHL and were admitted for high dose steroid therapy between January 2009 and December 2021. Patients were excluded if they had vertigo, chronic otitis media, a history of middle ear cavity surgery, disorders of the central nervous system, tumors, or progressive sensorineural hearing loss.

The medical records of the patients were reviewed, and their demographic and clinical characteristics were recorded, including age, sex, body mass index, medical history, comorbid symptoms, days from onset to treatment, time to recovery, and duration of recurrence. Only current alcohol use and smoking were considered, respectively. Following admission, all patients underwent daily pure tone audiometry (PTA) tests in a sound-treated audiology booth in the ENT clinic by qualified and experienced audiologists, with hearing thresholds calculated using the six-division method ((500 Hz + (1000 Hz × 2) + (2000 Hz × 2) + 4000 Hz)/6).

High dose steroid therapy consisted of 1 mg/kg/day oral methylprednisolone for four days, followed by tapering. Patients were discharged after tapering and followed in the outpatient clinic. Hearing threshold at recovery was defined as the threshold observed when PTA tests showed no improvement for two weeks. Hearing recovery was assessed using Siegel’s classification [13], with patients categorized as experiencing complete, partial, slight, and no recovery for both episodes of ISSNHL. Complete, partial, and slight recovery were grouped as the recovery group. Factors were compared in patients who did and did not recover after both episodes (Table 1). To analyze factors associated with poorer outcomes after the recurrent than after the first episode of ISSNHL, patients who recovered after the first episode were divided into those who experienced similar or better recovery and those who experienced poorer recovery after the recurrent than after the first episode. In this analysis, patients who did not recover after the first episode were excluded (Table 2).

### 2.2. Statistical Analysis

Categorical variables were compared using Fisher’s exact tests and continuous variables were compared using Wilcoxon rank sum tests. Factors associated with recovery were assessed by multiple ordinal logistic regression analysis, adjusted by age and sex. All statistical analyses were performed using the SAS9.4 (Statistical Analysis System version, SAS Institute, Cary, NC, USA) software, with a *p*-value < 0.05 defined as statistically significant.

### 2.3. Research on Recurrent ISSNHL

This systematic review was performed in accordance with the Primary Reporting Items for Systematic Review and Meta-analyses (PRISMA) statement [14]. Studies were included if (a) they were retrospective or prospective investigative studies; (b) all patients had been diagnosed with recurrent idiopathic sudden sensorineural hearing loss, with those having other causes of hearing loss excluded; and (c) the studies were published in English. Studies were excluded if they were (a) unpublished data; (b) review articles; (c) grey literature; (d) case reports; or (e) duplicates of published research. Studies published before 1 September 2021 were retrieved by one of the investigators (S.Y.J.) from three electronic databases: SCOPUS, PubMed and the Cochrane Library. Search terms included “recurrent sudden hearing loss”.

## 3. Results

During the study period, 38 patients with recurrent ISSNHL were treated for recurrent ISSNHL; their demographic, audiometric, and other clinical characteristics are summarized in Table 1. The 38 patients consisted of 14 (36.8%) men and 24 (63.2%) women. Their mean age at the first episode was 48.9 years (range 7–69 years), and their mean age at the recurrent episode was 52.7 years (range 12–76 years). Recurrences were ipsilateral in thirty-five (92.1%) patients and contralateral in three (7.9%). Thirty (78.9%) patients recovered after the first episode, whereas eight (21.1%) did not. In comparison, 24 (63.2%) patients recovered after the recurrent episode, whereas 14 (36.8%) did not.

A comparison of demographic and clinical characteristics of the patients who did and did not recover after the first episode showed no significant differences. In contrast, a comparison of patients who did and did not recover after the recurrent episode showed that the symptom of ear fullness (91.7% vs. 28.6%; *p* < 0.05) was significantly more frequent, time to treatment onset significantly shorter (5.6 days vs. 16.7 days, *p* < 0.05) and recovery time significantly shorter (1.3 months vs. 3.1 months, *p* < 0.05) in patients who did rather than did not recover after the recurrent episode (Table 1).

Of the 30 patients who recovered after the first episode, 18 (60.0%) showed better or similar recovery after the recurrent episode, whereas 12 (40.0%) showed poorer recovery after the recurrent rather than after the first episode (Table 3). Ear fullness symptoms were significantly less frequent (50% vs. 88.9%, *p* < 0.05) and mean initial hearing threshold significantly higher (57.2 dB vs. 40.1 dB, *p* < 0.05) in patients who did rather than did not experience poorer recovery after the recurrent rather than after the first episode of ISSNHL (Table 3).

Based on the results of preliminary simple linear regression analysis to determine the factors associated with poorer recovery after the recurrent rather than after the first episode of ISSNHL, multiple regression analysis was performed with adjustment for age and sex (Table 4). Ear fullness symptoms at the time of ISSNHL recurrence were associated with a lower risk of poorer recovery after the recurrent episode (*p* < 0.05, odds ratio (OR) 0.1, 95% confidence interval (CI)] 0.01–0.76). Higher hearing loss at the time of ISSNHL recurrence was associated with a higher risk of poorer recovery after the recurrent episode (*p* < 0.05, OR 1.06, 95% CI 1.01–1.12).

Following ISSNHL recurrence, 23 patients underwent temporal bone MRI scans to rule out brain tumors or other ear abnormalities. None of these patients showed evidence of cerebellopontine angle tumors.

Using the search term “recurrent sudden hearing loss”, 35 papers were identified from the three databases. Only nine (25.7%) of these studies fulfilled the inclusion criteria and were reviewed and these are shown in Figure 1. The characteristics of the included studies are shown in Table 5. This review especially focused on recovery rate from recurrent ISSNHL, although two of these studies [7,8] only provided data on the recurrence rate. The nine previous studies reported that the rate of recovery in patients with recurrent ISSNHL ranged from 43.4% to 78.6%. Only one study reported the second recurrence rate, which was 21.4%. In most studies, the proportion of females and males was almost the same. These studies utilized different tests, including electrocochleography (ECohG) and vestibular-evoked myogenic potential (VEMP), and analyzed different parameters, including neutrophil to lymphocyte ratio (NLR) and platelet to lymphocyte ratio (PLR), to identify prognostic factors associated with recurrent ISSNHL.

**Table 5 jcm-11-01453-t005:** Studies assessing recurrent ISSNHL.

Reference	Country	Study Design	Number of Patients with Recurrence	Recovery Rate from Recurrence	M:F	Conclusions
Seo et al. [9]	South Korea	Retrospective	First recurrence: 16 Second recurrence: 16	First recurrence: 78.6% Second recurrence: 21.4%	8:8 8:8	NLR and PLR higher in patients with both recurrent and non-recurrent ISSNHL.
Park et al. [5]	South Korea	Retrospective	11	72.7%	6:5	Hearing outcomes were poorer after a recurrent than after the first episode, with SSNHL almost always recurring in the same ear.
Ohashi et al. [6]	Japan	Retrospective	23	69.5%	NA	Favorable prognostic factors in patients with recurrent ISSNHL included an enhanced SP/AP ratio of ECohG, a low initial AP threshold, a low initial hearing level, and an up-sloping type of audiogram. Initial vertigo was associated with unfavorable outcomes in patients with recurrent ISSNHL.
Kuo et al. [15]	Taiwan	Retrospective	Ipsilateral: 7 Contralateral: 9	50% (Ipsilateral type: 71.4% Contralateral type: 33.3%)	3:4 5:4	Normal VEMPs in the affected ear of patients with recurrent sudden deafness may indicate a good hearing outcome.
^†^ Fushiki et al. [7]	Japan	Retrospective	33	* Recurrence rate 1 year: 29% 5 years: 47% (45% of recurrences occurred within 6 months of the first episode)	-	Recurrence rate higher in patients with elevated SP/AP and spontaneous nystagmus (78.6%) than in patients with normal SP/AP and absence of spontaneous nystagmus (31.8%)
Furuhashi et al. [3]	Japan	Retrospective	14	78.5%	9:5	Recurrence of sudden deafness rare during long-term follow-up The degree of hearing deterioration on the first affected side was not significantly different from that on the non-affected side
^†^ Wu et al. [8]	Taiwan	Retrospective	2281 (Data from the Taiwan NHI)	* Recurrence rate 5 years: 4.99%	1252:1029	Factors associated with relapse included age 35–64 years, diabetes mellitus, and hypercholesterolemia
Pecorari et al. [16]	Italy	Retrospective	73	63% * Recurrence rate 2 years: 5.6% 5 years: 10.34%	30:43	Recurrence correlated only with the presence of tinnitus during follow-up
Wu et al. [17]	Taiwan	Retrospective	30	43.44% (First episode: 53.55%)	16:14	Hearing recovery after a recurrent episode correlated significantly with hearing outcome after the initial episode.

Abbreviation: NLR, neutrophil to lymphocyte ratio; PLR, platelet to lymphocyte ratio; ISSNH, idiopathic sudden sensorineural hearing loss; SP, summating potential; AP, action potential; ECohG, electrocochleography; VEMP, vestibular-evoked myogenic potential; NHI, National Health Insurance program * Recurrence rate: cumulative recurrence rate. ^†^ These studies only reported about relapse rate.

## 4. Discussion

Many patients who experience recurrent ISSNHL report significant stress relative to the recurrence and are more concerned about recovery than during the first episode of ISSNHL. Clinicians also express concern about the likelihood of recovery relative to the first episode and about the possibility that recurrent ISSNHL is caused by a tumor.

Most cases of SSNHL are idiopathic; other causes, however, can include vestibular schwannoma, acoustic neuroma, stroke, malignancy, Meniere’s disease, trauma, autoimmune disease, syphilis, Lyme disease, and peri-lymphatic fistula [1,18]. The prevalence of vestibular schwannomas in patients with SSNHL has been reported to be 3.0% [19].

This study assessed 38 patients with recurrent ISSNHL, focusing on the factors associated with recovery. The control group of the present study consisted of patients who did not recover from recurrent ISSNHL. If the process of relapse consists of a series of clinical steps, then the factors among patients who recovered after the first episode may differ between those who did and did not experience poorer recovery after the recurrent episode. ISSNHL is an uncommon disease, and recurrent ISSNHL is very rare; hence, knowledge about the factors associated with patient recovery from recurrent episodes may help in counseling and treating patients.

In general, patients with ISSNHL are advised to start treatment with oral steroids within two weeks [1], and starting treatment after 10 days is a negative prognostic factor [20]. Among patients who experienced recurrent episodes in the present study, those who recovered started their treatment at a mean 5.6 days, whereas those who did not recover started their treatment at a mean 16.7 days, further indicating that delayed treatment was associated with a decreased recovery rate (*p* < 0.05). During the first episode, however, the mean time to treatment onset was similar in patients who did and did not recover (7.2 days vs. 5.5 days). Patients with previous experience of ISSNHL may delay visits to the hospital because they become empirically accustomed to symptoms such as sudden ear fullness, tinnitus, and hearing loss. This may delay the diagnosis and treatment of recurrent ISSNHL. The present findings indicate that patients who experience symptoms of recurrent ISSNHL should seek treatment as soon as possible.

The present study also showed that the ear fullness symptoms and the initial hearing level of the affected ear differed significantly between patients with and without poorer recovery after the recurrent episode than after the first episode of ISSNHL. Recovery of hearing after the first episode was shown to be prognostic for the subsequent hearing outcomes after the recurrent episode [17]. The present study found that the initial hearing loss of the affected ear at the time of recurrence was significantly higher in poorer recovery after the initial episode, as well as being a risk factor for poorer recovery after the recurrent than after the first episode (OR 1.06, 95% CI 1.01–1.12, *p* < 0.05) after adjustment for age and sex. Initial profound hearing loss was previously shown to be negatively prognostic for recovery from non-recurrent ISSNHL [20]. Similarly, the present study found that higher initial hearing loss at the time of the recurrent episode was prognostic for poorer recovery after the recurrent than after the first episode.

Ear fullness was experienced by six (50.0%) of the twelve patients who experienced poorer recovery after recurrence than after initial ISSNHL and 16 (88.9%) of the 18 patients who did not. Ear fullness is a subjective symptom, expressed as feelings of blockage, plugging, or pressure, that occurs frequently in patients with acute sensorineural hearing loss. For example, the incidence of ear fullness has been reported to be 63.5% in patients with acute low tone sensorineural hearing loss [18], 40.2% in patients with ISSNHL [1], and 61.0% in patients with Meniere’s disease [21]. Ear fullness in acute sensorineural hearing loss was not associated with auditory function on audiograms but was associated with the low-frequency region [21]. This association disappeared after the hearing threshold stabilized, and the disappearance of ear fullness has been associated with hearing prognosis. Ear fullness observed in patients with endolymphatic hydrops or Meniere’s disease has been associated with a pressure imbalance between the round and oval windows [17,22,23]. Although the mechanisms underlying these associations have not yet been determined, these findings suggest that ear fullness may be due to a functional factor rather than to an anatomical impairment of the cochlea. The mechanism responsible for ear fullness in patients with sensorineural hearing loss needs to be clarified [21,24].

In this study, patients who showed better or similar recovery from the recurrent rather than from the first episode of ISSNHL had more frequent symptoms of ear fullness than those who showed poorer recovery from recurrent ISSNHL. Although the ear fullness symptom was associated with the low-frequency region, however, none of the six patients with poorer recovery after the recurrent episode and only four of the sixteen patients without poorer recovery after the recurrent episode who presented with ear fullness showed a low tone loss pattern, indicating that ear fullness symptom is not present only in patients with low tone sudden sensorineural hearing loss. In some respects, patients with poorer recovery after the recurrent episode showed a higher initial hearing loss and more frequent ear fullness symptom at the same time than did patients without poorer recovery after the recurrent episode, suggesting that ear fullness affected the remaining or early stages of loss of hearing function. Taken together, these findings suggest that ear fullness may be prognostic of recovery from recurrent ISSNHL.

ISSNHL has been reported to be associated with a history of hypertension (HTN), chronic kidney disease, diabetes mellitus (DM), hypercholesterolemia, and stroke [25,26,27,28,29]. Analysis of data from the US National Nutrition Survey showed a link between hearing loss and high blood pressure [30]. HTN can cause hemorrhage in the inner ear, reducing capillary blood flow and oxygen supply, and resulting in progressive or sudden sensorineural hearing loss [31]. Moreover, older age and HTN are significant negative prognostic factors for recovery from ISSNHL [20,32]. In the present study, however, HTN was not a statistically significant risk factor for poorer recovery after a recurrent rather than after the first episode of ISSNHL. However, our finding, that HTN was more frequent in patients with poorer recovery after a recurrent than after a first episode, suggests that HTN may be a risk factor for poorer recovery from recurrent ISSNHL. Moreover, acquired and inherited cardiovascular risk factors were found to be associated with an increased risk of developing ISSNHL [33]. The management of cardiovascular diseases, including HTN, is therefore important in preventing the recurrence of ISSNHL.

The rates of DM in patients showing better or similar recovery compared to poorer recovery after a recurrent episode rather than after a first episode were 22.22% and 25.00%, respectively. Several studies have reported an association between ISSNHL and DM [25,26,27,28,29], although one previous study [34] found no difference in the outer hair cell (OHC) damage, represented by evoked otoacoustic emissions (e-OAEs), between DM patients and healthy subjects in an acute hyperglycemic clamp study which is assumed as a sudden hazardous condition. Sensorineural hearing loss in DM patients has been shown to be associated with longer duration and poor control of DM; however, a higher compromise is observed in DM patients in an acute hyperglycemia condition [34], so it is hard to find the correlation between the acute hyperglycemia condition and ISSNHL. However, it is clear that DM has an effect on hearing loss in the long term, and must be managed appropriately.

The recovery rate of recurrent ISSNHL of our study was 63.2% and it ranged in between the previous studies in the systematic review (43.4~78.6%). Similar to our study, hearing outcomes were found to be poorer after the recurrent rather than after the initial episode [5,17]. A previous study reported that 45% of recurrences occurred within 6 months from the first episode [7]. In comparison, the present study found that the mean times to recurrence were 43.2 days in patients who recovered from the recurrent episode and 44.9 days in those who did not recover. The cumulative relapse rates in studies included in the systematic review ranged from 4.9% to 47% [7,8,16].

Those studies found that statistically significant factors associated with recurrence included diabetes mellitus, hypercholesterolemia, and tinnitus. ISSNHL has also been associated with a history of HTN, chronic kidney disease, and stroke. Although the present study found that HTN was not a statistically significant risk factor for recurrent ISSNHL, its prevalence was higher in patients with than without poorer recovery after the recurrent rather than after the first episode of ISSNHL. Both the neutrophil to lymphocyte ratio (NLR) and platelet to lymphocyte ratio (PLR) were found to be higher in patients experiencing both the first and recurrent episodes of ISSNHL than in a normal control group [9]. Similar to our findings, electrocochleography (ECohG) showed that low initial hearing level is prognostic for recovery from ISSNHL [6]. Moreover, a more normal vestibular-evoked myogenic potential (VEMP) [15] and better hearing outcome after the initial episode [17] were found to be associated with better hearing outcomes in patients with recurrent ISSNHL.

Although few studies to date have assessed prognosis in patients with recurrent ISSNHL, various clinical, serologic, and audiologic factors have been analyzed. Large population studies have failed to identify factors that are prognostic or predictive of the degree of recovery from recurrent ISSNHL, but in general the factors associated with recurrent ISSNHL have been found not to differ significantly from the factors associated with primary ISNSHL. The present study found that the degree of recovery was associated with treatment onset. A lower initial hearing level and presence of ear fullness symptom were associated with better recovery from the recurrent than from the first episode of ISSNHL. Although further research is needed on the mechanisms responsible for ear fullness, ear fullness can likely be recognized during early stages of functional loss, allowing earlier treatment and improved prognosis.

This study had several limitations. First, this study only analyzed patients who experienced recurrent ISSNHL. To more comprehensively analyze factors prognostic of recurrent ISSNHL, the characteristics of patients who experienced ISSNHL once and those who experienced recurrent ISSNHL should be compared. Second, this study was retrospective in design, which may have introduced a potential selection bias. Third, longer-term serial follow up of patients after recovery from recurrent ISSNHL is needed to specifically classify the patients who might experience additional recurrence. Fourth, a small number of subjects were studied and there was a higher proportion of females compared to males.

## 5. Conclusions

Earlier treatment in patients with recurrent ISSNHL is associated with better hearing outcomes. In addition, ear fullness was associated with a more favorable prognosis, whereas higher initial hearing loss was predictive of a poorer recovery after recurrent than after initial ISSNHL.

## Figures and Tables

**Figure 1 jcm-11-01453-f001:**
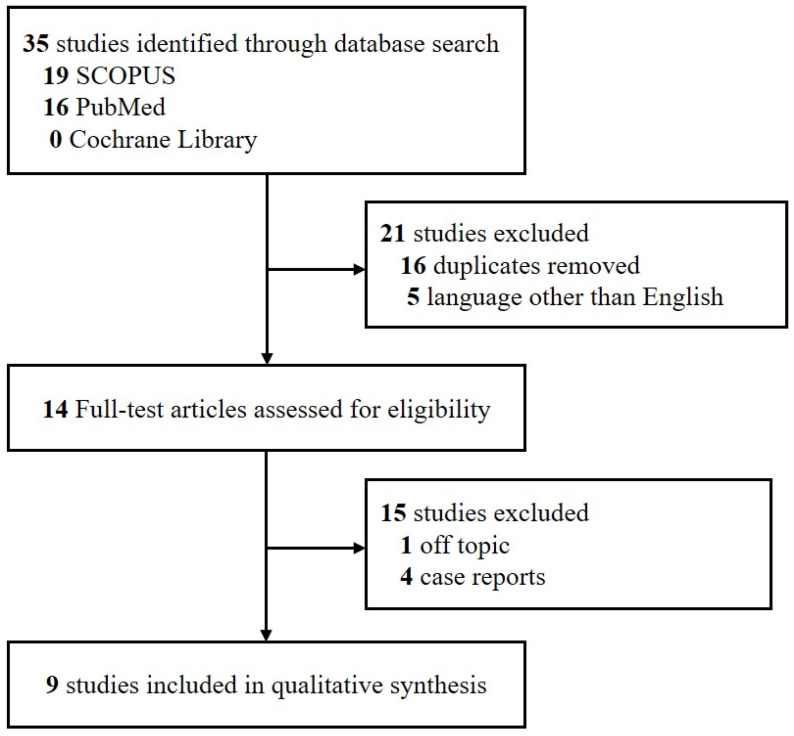
PRISMA flow diagram. Abbreviation: PRISMA: Primary Reporting Items for Systematic Review and Meta-analyses.

**Table 1 jcm-11-01453-t001:** Characteristics of patients who did and not recover after the first and recurrent episodes of idiopathic sudden sensorineural hearing loss.

Variables		First Episode			Recurrent Episode		
		Recovery (*n* = 30)	No Recovery (*n* = 8)	*p*-Value	Recovery (*n* = 24)	No Recovery (*n* = 14)	*p*-Value
		*n* (%) or Mean ± SD	*n* (%) or Mean ± SD		*n* (%) or Mean ± SD	*n* (%) or Mean ± SD	
Age (year) mean ± SD		48.90 ± 16.40	48.75 ± 18.11	0.7489	53.63 ± 14.13	51.00 ± 19.52	0.9880
Sex	Male	12 (40.00%)	2 (25.00%)	0.6836	8 (3.33%)	6 (42.86%)	0.7293
	Female	18 (60.00%)	6 (75.00%)		16 (66.67%)	8 (57.14%)	
BMI (kg/m^2^), mean ± SD		22.89 ± 2.57	22.42 ± 2.78	0.6605	22.78 ± 2.75	22.80 ± 2.37	0.8373
Alcohol		4 (13.33%)	3 (37.50%)	0.1461	5 (20.83%)	2 (14.29%)	1.0000
Smoking		8 (26.67%)	2 (25.00%)	1.0000	5 (20.83%)	5 (35.71%)	0.4485
HTN		5 (16.67%)	2 (25.00%)	0.6236	3 (12.50%)	4 (28.57%)	0.3870
DM		7 (23.33%)	1 (12.50%)	0.6600	6 (25.00%)	2 (14.29%)	0.6836
Tinnitus		19 (63.33%)	6 (75.00%)	0.6893	16 (66.67%)	12 (85.71%)	0.2685
Ear fullness		20 (66.67%)	8 (100.00%)	0.0821	22 (91.67%)	4 (28.57%)	0.0001 *
Treatment onset (days), mean ± SD		7.17 ± 16.07	5.50 ± 5.01	0.5193	5.58 ± 11.99	16.71 ± 20.73	0.0361 *
Recovery time (months), mean ± SD		0.95 ± 1.04	4.57 ± 10.29	0.2630	1.31 ± 2.04	3.01 ± 5.07	0.0303 *
Hearing level of the affected ear before treatment (dB), mean ± SD		48.44 ± 25.45	47.60 ± 16.27	0.8588	44.58 ± 21.94	49.17 ± 11.92	0.5686
Hearing level of the affected ear after treatment (dB), mean ± SD		21.00 ± 14.87	41.56 ± 15.88	0.0023 *	23.13 ± 13.60	53.39 ± 15.35	<0.0001 *
Time to recurrence (days), mean ± SD		-	-	-	43.22 ± 54.31	44.86 ± 37.84	0.3697

Abbreviation SD, standard deviation; BMI, body mass index; HTN, hypertension; DM, Diabetes mellitus. * *p* < 0.05.

**Table 2 jcm-11-01453-t002:** Distribution of patients according to types of recovery after the first and recurrent episodes of idiopathic sudden sensorineural hearing loss. Complete, partial, slight and no recovery were determined according to the Siegel’s criteria [13].

		Recurrent Episode				
	Recovery Type	Complete	Partial	Slight	No	Total
**First episode**	Complete	15	0	1	5	21
	Partial	1	2	2	3	8
	Slight	0	0	0	1	1
	No	2	1	0	5	8
	Total	18	3	3	14	38

**Table 3 jcm-11-01453-t003:** Characteristics of patients who did and did not experience a poorer recovery after a recurrent episode than after the first episode of idiopathic sudden sensorineural hearing loss.

Variables		No Worse than after the First Episode (*n* = 18)	Worse than after the First Episode (*n* = 12)	*p*-Value
		*n* (%) or Mean ± SD	*n* (%) or Mean ± SD	
Age (year), mean ± SD		52.33 ± 13.81	53.00 ± 20.78	0.5855
Sex	Male	7 (38.89%)	5 (41.67%)	1.0000
	Female	11 (61.11%)	7 (58.33%)	
BMI (kg/m^2^), mean ± SD		22.70 ± 2.71	23.18 ± 2.42	0.3645
Alcohol		3 (16.67%)	1(8.33%)	0.6315
Smoking		5 (27.78%)	3 (25.00%)	1.0000
HTN		1 (5.65%)	4 (33.33%)	0.1282
DM		4 (22.22%)	3 (25.00%)	1.0000
Tinnitus		12 (66.67%)	10 (83.33%)	0.4192
Ear fullness		16 (88.89%)	6 (50.00%)	0.0342 *
Treatment onset (days), mean ± SD		5.67 ± 13.69	8.33 ± 10.74	0.1721
Recovery time (months), mean ± SD		1.26 ± 2.25	1.47 ± 1.29	0.0992
Hearing level of the affected ear before treatment (dB), mean ± SD		40.05 ± 20.17	57.22 ± 12.92	0.0357 *
Hearing level of the affected ear after treatment (dB), mean ± SD		18.29 ± 8.47	52.36 ± 12.08	<0.0001 *
Time to recurrence (days), mean ± SD		34.65 ± 37.54	53.35 ± 37.46	0.1139

Abbreviation SD; standard deviation, BMI; body mass index, HTN; hypertension, DM; diabetes mellitus. * *p* < 0.05.

**Table 4 jcm-11-01453-t004:** Adjusted risk factors for poorer recovery after a recurrent rather than after the first episode of ISSNHL.

Variables	Simple Logistic Model			Multiple Logistic Model *		
	OR	95% CI	*p*-Value	OR	95% CI	*p*-Value
Age (year)	1.00	0.96–1.05	0.9128	adj.		
Female (ref. Male)	0.89	0.20–3.95	0.8791	adj.		
BMI	1.08	0.80–1.45	0.6213	1.01	0.69–1.47	0.9703
Alcohol	0.46	0.04–4.98	0.5185	0.44	0.04–4.89	0.4997
Smoking	0.87	0.16–4.58	0.8662	0.69	0.09–5.67	0.7323
HTN ^†^	6.18	0.69–55.18	0.1032	10.03	0.82–123.31	0.0717
DM	1.17	0.21–6.48	0.8602	1.17	0.17–8.2	0.8761
Tinnitus	2.50	0.41–15.23	0.3203	2.58	0.42–16.04	0.3092
Ear fullness	0.13	0.02–0.80	0.0280	0.10	0.01–0.76	0.0262 **
Treatment onset (days)	1.02	0.96–1.09	0.5032	1.03	0.96–1.11	0.4077
Recovery time (months)	1.06	0.72–1.56	0.7698	1.06	0.72–1.57	0.7546
Hearing level of the affected ear before treatment (dB)	1.06	1.01–1.11	0.0263	1.06	1.01–1.12	0.0210 **
Time to recurrence (days)	1.02	1.00–1.03	0.0442	1.02	0.99–1.04	0.1569

Abbreviation OR; odds ratio; BMI; body mass index; HTN; hypertension; DM; diabetes mellitus. ^†^ Logistic regression with Firth’s method (reference: patients not experiencing poorer outcomes after recurrence). * Logistic regression of patients with poorer outcomes after the recurrent than after the first episode (reference: patients not experiencing poorer outcomes after recurrence). * Adjusted by age, gender. ** *p* < 0.05.

## Data Availability

The data are available from authors by a reasonable request.

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
