# Peer review of "Prognostic Factors Associated with Recovery from Recurrent Idiopathic Sudden Sensorineural Hearing Loss: Retrospective Analysis and Systematic Review"

_jcm, 2022, doi:10.3390/jcm11051453_

Round 1
Reviewer 1 Report
This study offers a interesting description of potential factors for recovery after recurrent ISSNHL.
Strengths of the paper include a complete introduction, inclusion of some relevant literature references and a correct adherence to methodological structure of cross-sectional studies.
However, I would like to make some suggestions, and ask for some changes in order to improve the manuscript:
- Title: As this study includes both a retrospective study, and a systematic review, this should be included in the title
- Methods:
- I suggest to improve the section "Research on recurrent ISSNHL", and to perform a more systematic review of the literature, by developing its methodology through the PRISMA guidelines (search strategy, selection criteria, variable of interest, data synthesis...), and including a flowchart of the review process. In addition, these results should be included in the "Results section", and should be annotated in the "Discussion section". I know this
- There is a lack of "Limitations" section in the discussion. I can list some issues that should the included in this section (the potential inclusion of bias related to retrospective studies (selection bias...).
- I think it would be interesting to perform a comparison of the results obtained in the first ISSNHL episode with other risk factor described in isolated episodes of sudden SNHL, to describe differences between single ISSNHL episodes compared to recurrent episodes.
In addition, I've some minor questions, that can be easily addressed:
- This is not a request itself. Could you explain the advantages of the "six-division method" compared to the Pure tone average, to summarize the hearing thresholds?
- In the "methods section", the route of drug administration is missing: "High dose steroid therapy consisted of 1 mg/kg/day methylprednisolone for 4 days, followed by tapering."
- I suggest to change the way to express the worsening of hearing thresholds. For example: "Higher initial hearing threshold at the time of ISSNHL recurrence were associated with a higher risk of poorer recovery after the recurrent episode (p<0.05, OR 1.06, 95% CI 1.01-1.12)." should be changed to "Worst initial hearing threshold..." in order to make it clearer. This has been expressed this way through all the paper. Please, make this change in all the manuscript.
Reviewer 2 Report
- The authors use the terminology, in several places throughout the manuscript, of “higher initial hearing level” and “poorer initial hearing level”. They observe that “higher initial hearing level was prognostic for poor recovery”. This reviewer interprets this to mean that poorer initial hearing was a bad prognostic indicator. Is this correct? Please clarify this use of “higher and poorer hearing level” at its first use or consider less ambiguous language.
- Abstract, second to last sentence, “symptom” should be “symptoms”
- During admission, patients underwent daily pure tone audiometry. Was this done using portable, bedside audiometers in the ward setting or was this done in a sound-treated audiology booth in the clinic?
- The authors describe using Siegl’s classification for hearing recovery, where three groups are recognized – complete, partial, and slight recovery. Distribution of patients using this schema are shown in Table 2. Table 1, however, parses patients into two groups – recovery and nonrecovery. How are these two approaches reconciled? Are the investigators considering Siegl’s complete as “recovery” and Siegl’s partial and slight as “nonrecovery”? Or are they grouped differently? This should be made clear in the Methods section.
- For the systematic review portion, review inclusion and exclusion criteria need to be stated. It is stated that non-English, case reports, and “lack of relevant” data were exclusion criteria. Could the authors be more specific? What data were they specifically looking for to be considered for inclusion?
- The authors make a statement in the Discussion section that, “These findings suggest that ear fullness may be due to a functional factor rather than an organic lesion of the cochlea.” Can the authors please expand or clarify this statement? How do they conceptualize the difference?
- Please reword the sentence in Discussion for clarity, “However, our finding, that more patients with than without poorer recovery after a recurrent episode…”
- Table I. Alcohol and smoking are examined as behavioral factors. The authors should make clear in Methods whether is yes/no any current alcohol or smoking or some other history of these.
- Table I needs a legend where abbreviations are written out (e.g., HTN, DM, BMI, etc.)
- Same comment for Table II
- Table IV also needs abbreviations written out
- Could the authors please provide numbers to facilitate reference to the text in review?
Reviewer 3 Report
The study is interesting. The conclusions are supported by the results. The tables and figure are quite clear.
However, this reviewer raises some issues that the authors need to address.
1- The authors well point out that ISSNHL has been reported to be associated with diabetes mellitus. Indeed, sensorineural damage has been observed to be related to diabetes and the degree of metabolic control, expressed as HbA1c, but not related to acute hyperglycemia such as that induced during a hyperglycemic clamp in both diabetic and healthy subjects (Metabolism. 1999 Nov; 48(11):1346-50. doi: 10.1016/s0026-0495(99)90141-5.). This interesting issue and the above reference should be addressed in discussion.
2- The study has some weaknesses: the small number of subjects studied, the higher prevalence of the female population compared to the male one which does not allow to exclude a gender difference, etc.). Therefore, the authors must insert a section on the limitations of the study to the end of the discussion.
3- There is an error in table 1. About DM patients with recurrent episode and non-recovery, 2 out of 14 is not 85.71%. Please correct.
4- The paper should be reviewed by a native English speaker.
Round 2
Reviewer 3 Report
No further comments.